# A Novel Pigeon-Inspired Optimization Based MPPT Technique for PV Systems

**Ai-Qing Tian [1]**, **Shu-Chuan Chu [1,2]**, **Jeng-Shyang Pan [1,3,*]** and **Yongquan Liang [1]**

1. College of Computer Science and Engineering, Shandong University of Science and Technology, Qingdao 266590, China; stones12138@163.com (A.-Q.T.); scchu0803@gmail.com (S.-C.C.); lyq@sdust.edu.cn (Y.L.)
2. College of Science and Engineering, Flinders University, 1284 South Road, Clovelly Park SA 5042, Australia
3. Department of Intelligence Science and Technology, College of Informantion Science and Technology, Dalian Maritime University, Dalian 116026, China
* Correspondence: jspan@cc.kuas.edu.tw

**Abstract:** The conventional maximum power point tracking (MPPT) method fails in partially shaded conditions, because multiple peaks may appear on the power–voltage characteristic curve. The Pigeon-Inspired Optimization (PIO) algorithm is a new type of meta-heuristic algorithm. Aiming at this situation, this paper proposes a new type of algorithm that combines a new pigeon population algorithm named Parallel and Compact Pigeon-Inspired Optimization (PCPIO) with MPPT, which can solve the problem that MPPT cannot reach the near global maximum power point. This hybrid algorithm is fast, stable, and capable of globally optimizing the maximum power point tracking algorithm. Therefore, the purpose of this article is to study the performance of two optimization techniques. The two algorithms are Particle Swarm Algorithm (PSO) and improved pigeon algorithm. This paper first studies the mechanism of multi-peak output characteristics of photovoltaic arrays in complex environments, and then proposes a multi-peak MPPT algorithm based on a combination of an improved pigeon population algorithm and an incremental conductivity method. The improved pigeon algorithm is used to quickly locate near the maximum power point, and then the variable step size incremental method INC (incremental conductance) is used to accurately locate the maximum power point. A simulation was performed on Matlab/Simulink platform. The results prove that the method can achieve fast and accurate optimization under complex environmental conditions, effectively reduce power oscillations, enhance system stability, and achieve better control results.

**Keywords:** MPPT; Pigeon-Inspired Optimization; meta-heuristic algorithm; Particle Swarm Algorithm

## 1. Introduction

Recently, the human demand for energy is increasing. At present, the environmental pollution caused by traditional fossil energy is becoming increasingly serious. The end of fossil fuel use requires alternative sources of renewable energy. Solar energy has been widely used due to its universal, clean, huge, and long-lasting characteristics. A photovoltaic power generation system (PVPS) has the advantages of no harm to the environment, long-term operation, maintenance-free, etc., and is widely considered as an extremely attractive solution [1]. In addition, human beings are paying more and more attention to the protection of the environment. From global warming to the frequent and sudden increase in the number of haze days, people have been urged to continuously find and improve clean energy. It is imperative to gradually replace the high-carbon traditional fossil energy economic growth model with the low-carbon energy economic growth model.

First, the reserves of solar energy are quite abundant. About 130 trillion tons of solar energy converted into standard coal reaches the surface of the Earth every year, which is inexhaustible for human beings. Second, solar energy is a clean energy source that does not produce dust, carbon dioxide, or other toxic gases, nor does it cause environmental damage. At the same time, the equipment of photovoltaic power generation systems accounts for a very small part of the entire power generation system, but the growth rate has been very fast in recent years. In general, the development environment of photovoltaic power generation is quite excellent, and various factors have promoted the development of photovoltaic power generation. Therefore, the future power generation mode will take solar energy as the core of new energy-dominated power generation.

However, the current photovoltaic power generation equipment and algorithms are relatively dependent on the environment, and the utilization efficiency of solar energy is low, which will cause relatively large waste. The main reason is mainly due to the materials of photovoltaic cells and the control algorithms of photovoltaic power generation systems. However, the material performance of photovoltaic power generation is to absorb solar energy and convert it into electrical energy. To improve the material of photovoltaic cells, a large amount of manpower and time must be invested. Moreover, the effect of this improvement is not so obvious. Improving the MPPT algorithm in photovoltaic power generation systems has become the best way to significantly improve the efficiency of photovoltaic power generation systems.

The purpose of the intelligent calculation method is the same as the calculation method (or numerical analysis). Calculate a satisfactory approximate solution close to the real solution. Use this approximate solution instead of the real solution. In general, many problems have no analytical solution. At this time, calculation methods can be used to find numerical solutions. When the problem is really complicated, the calculation method is too large or it is difficult to implement the calculation method. In essence, intelligent computing methods are bionic, randomized, and empirical. Nature is random and empirical. It extracts this characteristic of nature and automatically adjusts to form experience.

At present, with the rapid development of computer technology [2–6], intelligent computing is one of the important methods of intelligent science, and it is also a cutting-edge subject of information technology. The intelligent computing technologies and new methods developed in recent years have been widely used in many disciplines, and have achieved fruitful results in military, financial engineering, nonlinear system optimization, knowledge engineering, and computer-aided medical diagnosis. Intelligent computing includes neural networks [7,8], fuzzy logic [9–11], and evolutionary computing [12]. Evolutionary computing is a type of random search optimization algorithm that simulates biological evolution and genetic principles (survival of the fittest). The evolution process of the entire group can be regarded as an optimization process, but the evolution trajectory of a single individual may not be an optimization process. Evolutionary calculations include Genetic Algorithm (GA) [13], Particle Swarm Optimization (PSO) [14–17], Grey Wolf Optimizer (GWO) [18–21], Cat Swarm Optimization [22–24], Differential Evolution (DE) [25–27], Ant Colony Optimization (ACO) [28–30], Artificial Bee Colony (ABC) [31,32], Flower Pollination Algorithm (FPA) [33,34], Bat Algorithm (BA) [35–37], QUasi-Affine TRansformation Evolutionary (QUATRE) [38–41], and Multi-Verse Optimizer (MVO) [42,43]. The pigeon swarm algorithm is a new meta-heuristic algorithm proposed by Duan in 2014 [44]. This algorithm is proposed to simulate the process of pigeons returning home. The algorithm includes two operator stages, which has strong global search ability and local convergence ability.

Although PIO was proposed recently, it has achieved a lot of research results in terms of model improvement and application. Starting from the introduction of the pigeon's autonomous homing behavior, the mechanism principle and mathematical model of the flock optimization algorithm are explained, and typical applications in the fields of drone formation [45], control parameter optimization [46], and image processing [47] are introduced.

## 2. Related Work

Compared with traditional power generation systems, the biggest difference of photovoltaic power generation systems is that the core component is photovoltaic cells. One is a thin-film battery. The conversion efficiency of this battery is about 13% or even lower. The other is a monocrystalline silicon and polycrystalline silicon photovoltaic cell. The conversion efficiency is higher than that of a thin-film battery, which can reach about 18–20%. Thin film batteries can be combined with buildings and are inexpensive, thus they have a wide range of applications.

### 2.1. Basic Principles of Photovoltaic Cells

The main material of photovoltaic cells is semiconductors, which have the best photovoltaic effect, that is, they are most efficient way to convert solar energy into electricity. This is because, when the PN node in the semiconductor is illuminated by sunlight, the neutral atoms in it will lose electrons and generate electron–holes. At this time, under the action of the electric field, the electrons will be forced to move toward the N region. The hole moves in the opposite direction to reach the P area. At this time, the PN junction generates a new electric field with opposite polarity at both ends, and has the opposite polarity to the original electric field. The interaction between the two forces the P zone to be positively charged and the N zone to be negatively charged. Photogenerated electromotive forces are generated at both ends. When a load is connected to both ends of the PN junction, a photogenerated current is generated, that is, electrical energy is available for the load. The photovoltaic effect principle diagram is shown in Figure 1.

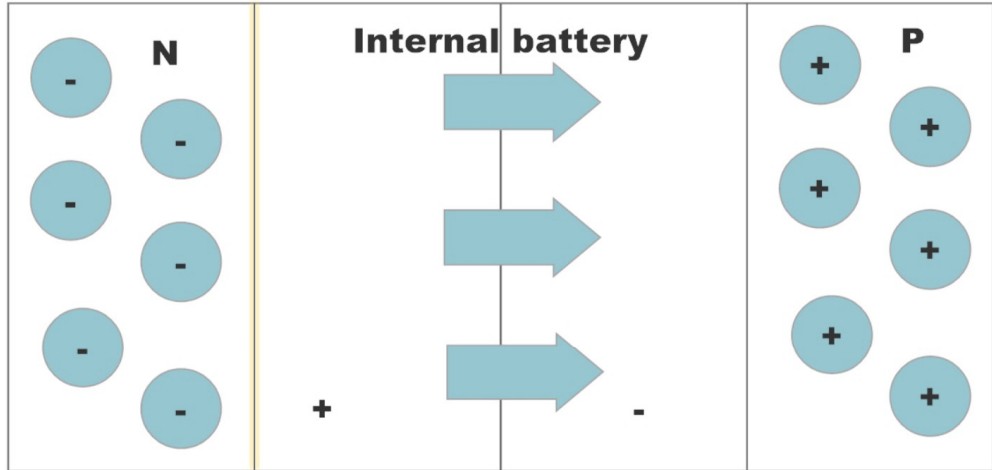

**Figure 1.** The principle diagram of the photovoltaic effect.

According to the description of the photovoltaic effect, solar light can be divided into three parts: one part is the light energy used by solar cells, in which photons are absorbed by the solar PN junction; and the other two parts are light energy that cannot be used, including the light energy reflected by the photovoltaic cell and the light energy absorbed by the other parts of the photovoltaic cell except the PN junction. Therefore, this is also a reason for the low efficiency of photovoltaic cells using solar energy one.

### 2.2. Physical Model of Photovoltaic Cell

Because the volt–ampere characteristics of photovoltaic cells are non-linear and easily affected by the external environment, physical models and mathematical models must be established to facilitate quantitative research. In fact, the mathematical model of photovoltaic cells is extremely complicated. In engineering applications, approximate equivalent circuits are often used instead. The approximate physical model of a photovoltaic cell is shown in Figure 2.

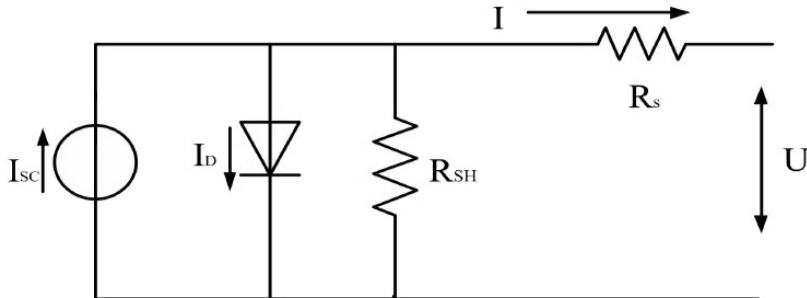

**Figure 2.** The physical approximate equivalent model of photovoltaic cells.

In Figure 2, the relationship between the port voltage of the solar cell and the current flowing to the port can be expressed by a mathematical equation, as shown in Equation (1).

$$
\begin{cases}
I_{PV} = I_{SC} - I_D[exp\left(\frac{V_{PV}+I_{PV}R_S}{nN_SV_T}\right) - 1] - \frac{V_{PV}+IR_S}{R_{SH}} \\
\\
V_T = m\frac{KT}{q}
\end{cases}
\tag{1}
$$

In Equation (1), $I_{PV}$ is the current flowing through the port, $I_{SC}$ is the current flowing to the port when the equivalent circuit is short-circuited, $I_D$ is the reverse saturation current of the photovoltaic cell, $V_{PV}$ is the port voltage of the solar cell, $R_S$ is the series resistance in the equivalent circuit, $R_{sh}$ is the parallel resistance in the equivalent circuit, $n$ is the ideal factor, $N_S$ is the number of photovoltaic modules in series, and $k$ is the Boltzmann constant. $T$ is the absolute temperature of the photovoltaic cell, $q$ is the amount of electronic charge in the photovoltaic cell, and $V_T$ is the voltage equivalent of temperature. Since $k$ and $q$ are constant values, $V_T$ can be known by getting $T$. For example, when the temperature T is 27 °C, $V_T$ is 26 mV.

Because the output voltage of a photovoltaic cell is generally only 0.5 V, it must be connected in series and parallel to form photovoltaic modules to increase the port voltage. Therefore, the mathematical model of photovoltaic array current is shown in Equation (2).

$$
I_{PV} = N_P I_{SC} - N_P I_D[exp\left[\frac{V_{PV} + i_{pv}R_S}{nN_SV_T}\right] - 1] - \frac{V + IR_S}{R_{SH}}
\tag{2}
$$

Among them, $N_P$ is the number of photovoltaic cells connected in parallel, and other parameters are the same as those in Equation (1). The mathematical model of the photovoltaic cell voltage is derived from the analysis of the equivalent model diagram, as shown in Equation (3).

$$
V_{PV} = \frac{N_S AKT}{q} ln(1 + \frac{N_P I_{SC} - I}{N_P I_D})
\tag{3}
$$

Therefore, the equation for photovoltaic cell output power can be obtained by multiplying Equations (2) and (3), as shown in Equation (4).

$$
P_{PV} = V_{PV} I_{PV} = N_P I_{SC} V_{PV} - N_P I_D V_{PV}[exp\frac{VPV + I_{PV}R_S}{nN_SV_T} - 1]
\tag{4}
$$

Because the unit magnitude of the parameters in the formula varies widely, to facilitate the calculation and analysis, the photovoltaic cell model can be simplified. In Equation (2), the magnitudes of the parallel resistance and the series resistance are greatly different. The unit magnitude of the parallel resistance is thousands of ohms, and the unit magnitude of the series resistance is milliohms. Thus, Equation (2) is simplified as shown in Equation (5).

$$P_{PV} = N_P I_{SC} - I_D exp[\frac{q(V_{PV} + IR_S)}{N_S AKT}] + I_D \tag{5}$$

Considering that the multiples of the diode's reverse saturation current are exponentially distributed, the single diode reverse saturation current in Equation (5) can be ignored, thus it is finally simplified to Equation (6).

$$P_{PV} = N_P I_{SC} - I_D exp[\frac{q(V_{PV} + IR_S)}{N_S AKT}] \tag{6}$$

*2.3. Maximum Power Point Tracking*

When the photovoltaic power generation system is at a constant temperature and constant light intensity, the volt–ampere characteristics of the photovoltaic array remain basically unchanged, but are still non-linear, and the *P–U* characteristic curve always has a maximum value. When the temperature or light intensity changes, or the photovoltaic array is under complex lighting conditions, this maximum value starts to shift. At this time, the MPPT algorithm starts to work. It monitors the photovoltaic array voltage and current while changing the DC voltage control system and output power to verify that the output power is maximum.

According to the circuit theory, it can be concluded that, when the output impedance of the photovoltaic array and the impedance of the external load are the same, the output power of the photovoltaic array is the largest.

Among MPPT algorithms, the most widely used algorithm in engineering today is the Perturb and Observe algorithms (*P&O*), which is also one of the earliest algorithms. Its advantages are simple and efficient under non-complex lighting conditions, low cost, and convenient detection and calculation. The so-called disturbance is to increase or decrease a voltage increment *U* from the original detected voltage. Then, the principle of *P&O* is to obtain the current and voltage values at the current time through the current and voltage detection elements, calculate the power value at the current time, and then add a perturbation to the original voltage. If it is large, then *P&O* continues to increase the perturbation comparison, otherwise it reduces one perturbation. In short, the direction of the disturbance is consistent with the direction of power increase. The flowchart is shown in Figure 3.

*P&O* is the most commonly used MPPT algorithm today because of its simplicity and versatility. This is based on the fact that the derivative of power versus voltage is zero at the maximum power point. At a working point on the *P–V* curve, if the operating voltage of the photovoltaic array is disturbed in a given direction and $dP > 0$, then the disturbance is known to move the operating point of the array to MPP. Then, the *P&O* algorithm will continue to disturb the PV array voltage in the same direction. If $dP < 0$, the change of the operating point moves the photovoltaic array away from the MPP, and the *P&O* algorithm reverses the direction of the disturbance. This paper uses a five-parameter model of a single diode to reproduce the non-linear characteristics of a photovoltaic array. Considering any point on the I–V curve, a mathematical expression by which *P&O* determines the next disturbance direction can be written.

$$\delta PO = \partial P / \partial V \tag{7}$$

While *P&O* decides the direction of the next perturbation based on the sign of $\delta PO$, the discrete form of Equation (7) becomes Equation (8).

$$\Phi PO = \frac{P_k - P_{k-1}}{V_k - V_{k-1}} = \frac{\Delta P}{\Delta V} \tag{8}$$

where $\Phi PO$ is the discrete form of $\delta PO$, $P_k = I_k V_k$, and $P_{k-1} = I_{k-1} V_{k-1}$. The other notations have the following meaning: $P_k$, $V_k$, and $I_k$ are the power, voltage, and current at the *k*th (actual) sampling instance, respectively; and $P_{k-1}$, $V_{k-1}$, and $I_{k-1}$ are the power, voltage, and current at the previous sampling instance, respectively.

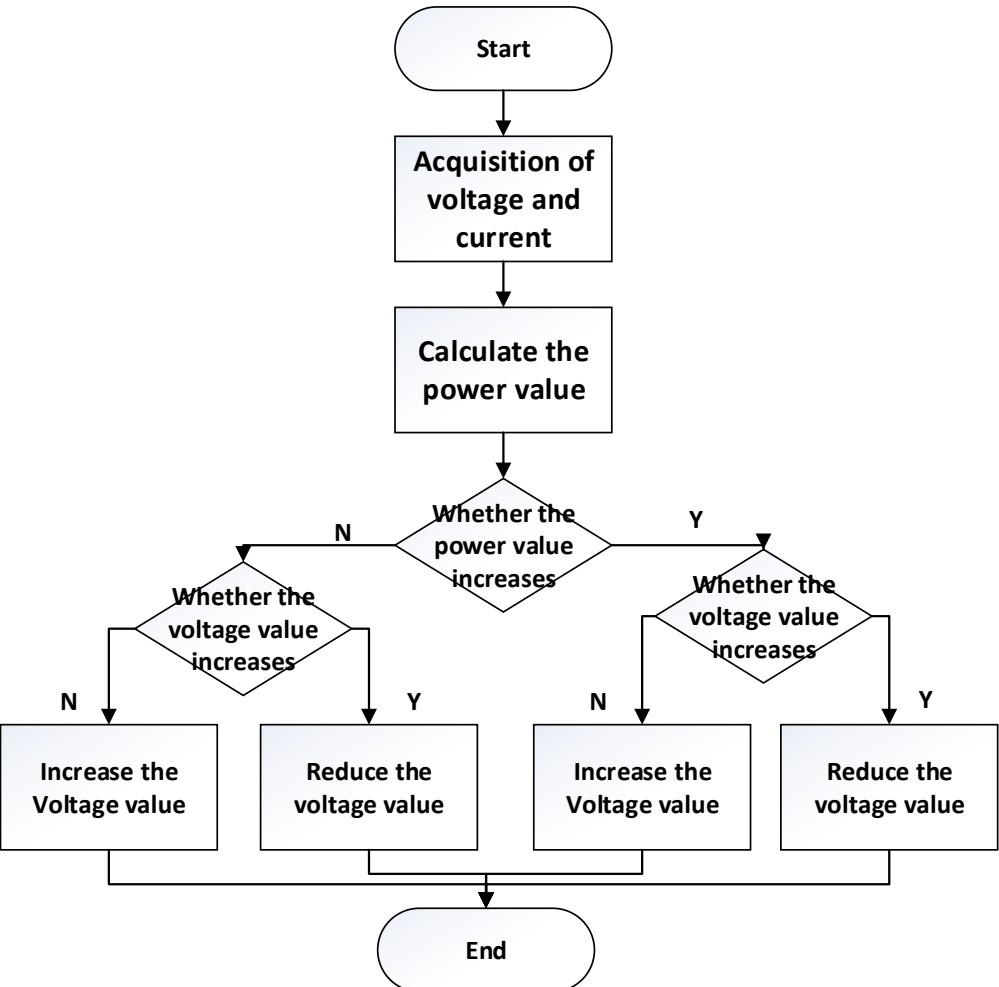

**Figure 3.** Flowchart of perturbation and observation algorithm.

However, according to the principle of *P&O*, the system will sway back and forth near the maximum power point, i.e., oscillating, causing power loss. In addition, because the perturbation of the *P&O* algorithm is a fixed value, at the beginning of the algorithm, the optimization speed is slow and cannot be changed, thus the power lost in the optimization process is large. To solve the problem that the optimization speed is slow and cannot be changed, a variable step size perturbation observation method has appeared.

## 3. Pigeon-Inspired Optimization

The Pigeon-inspired optimization algorithm is a new meta-heuristic algorithm. PIO was proposed in 2014 and was inspired by the behavior of pigeons returning home [48]. Pigeons can find their home with tools that help them to return home, the tools include magnetic fields, the sun, and landmarks. A magnetic field is formed in the pigeon's mind. The magnetic field can be used to shape the map in the pigeon's mind, and adjust its flight direction according to the height o and the angle of the sun. It will also affect the direction of the entire population. In the second phase, nearby landmarks help pigeons fly closer to their destination. To simulate the natural phenomenon of pigeons going home, the PIO algorithm uses two operators to describe the clustering behavior of carrier pigeons. In the PIO algorithm, the map and compass operators in the first stage represent the influence of the magnetic field and the sun, while the landmark operators in the second stage describe the influence of the landmark on the return of the pigeon.

### 3.1. Map and Compass Operator

Initialize the entire flock, the number of pigeons is $N_p$, the dimension of the activity is $D_{im}$, the map and compass factor are $R$, and the position and speed of the pigeons are expressed by Equations (9) and (10).

$$Pos_i = [Pos_{i1}, Pos_{i2}, ... Pos_{iD_{im}}] \tag{9}$$

$$Vel_i = [Vel_{i1}, Vel_{i2}, ... Vel_{iD_{im}}] \tag{10}$$

In the map and compass operator stages, the entire pigeon population presents a strong global search ability, which effectively avoids falling into a local optimum. The new position and speed update strategy for the flock is the following Equations (11) and (12).

$$Pos_i^{t+1} = Pos_i^t + Vel_k^{t+1} \tag{11}$$

$$Vel_i^{t+1} = e^{-R*t} * Vel_i^t + \phi_1 * (Pos_{gbest} - Pos_i) \tag{12}$$

In Equation (12), $Pos_{gbest}$ represents the position of the pigeon with the best fitness value in the entire population after each update, $t$ represents the number of iterations of the entire population, and $\phi_1$ is a variable limited to 0–1.

$$Vel^{t+1} = w * Vel_t + c1 * rand(0, 1)(P_{best} - Pos) + c2 * rand(0, 1)(G_{best} - Pos) \tag{13}$$

The speed update formula of the PSO algorithm is Equation (13), which is the migration direction of the simulated bird swarm. The inertial weight is introduced, and the inertial weight is adjusted linearly (or non-linearly) according to the process and the flight of the particles to balance the search. The speed update formula of PIO is Equation (12), which simulates the behavior of pigeons returning home. This formula is the speed update formula of the first stage. Unlike PSO, it has no individual optimal impact and has a strong global search ability to avoid the problem of getting trapped in a local optimum.

### 3.2. Landmark Operator

As the flock is approaching or nearing its end, the influence of maps and compass operators on the returning behavior of the entire flock becomes smaller. The target closer to the destination will become the new navigation reference for the pigeons until the pigeons return home. The PIO algorithm simulates this natural phenomenon using the pigeon with the best fitness value in the entire population as the center of the entire population, and the person with the most sensible way to lead the entire population to iteratively update. First, the entire pigeon population should be excluded from the pigeons that do not have a way to identify it to prevent these pigeons from affecting the iteration direction of the entire population, as shown in Equation (14). Second, choose the pigeon that has the most leadership in the entire population, as shown in Equation (15). Finally, this pigeon that leads the entire population is iterated, as shown in Equation (16).

$$N_p^t = \frac{N_p^{t-1}}{2} \tag{14}$$

$$Pos_{center}^t = \frac{\sum_{i=1}^{N_p^t} Pos_i^t * F(Pos_i^t)}{N_p^t * \sum_{i=1}^{N_p^t} F(Pos_i^t)} \tag{15}$$

$$Pos_i^{t+1} = Pos_i^t + \phi_4 * (Pos_{center}^t - Pos_i^t) \tag{16}$$

In Equation (15), $F(Pos_i^t)$ is the process of fitness value solution. When the function takes the maximum value and the minimum value, it has different expressions, as shown by Equation (17), $\varepsilon$ is a random constant that prevents $F(Pos_i^t)$ from being set to zero.

$$F(Pos_i^t) = \begin{cases} fitness(Pos_i^t) \ for \ maximization \ problem \\ \frac{1}{fitness(Pos_i^t)+\varepsilon} \ for \ minimization \ problem \end{cases} \qquad (17)$$

## 4. Hybrid Parallel and Compact Pigeon-Inspired Optimization

Since photovoltaic power generation cannot reach the maximum power under cover, this paper proposes an improved PIO algorithm based on parallel and compact hybrid. The improved PIO algorithm is a progress of previous work [19,20], considering both parallel and compact technologies. Parallel communication strategy is used to exchange information with other groups, share computing load, and enhance the calculation of individual diversity is of great significance. Compact technology can provide a very effective way to save variable memory. Taking advantage of the population-based algorithm, an effective method is adopted to save the variable memory without the need to store the actual population solution. This paper makes use of the advantages of this population-based algorithm, and does not need to store the actual population solution, using an effective method to represent the search space solution. The compact algorithm simulates the behavior of a population-based algorithm by replacing the virtual population with a probability representation.

### 4.1. Parallelized Pigeon-Inspired Optimization

In meta-heuristic algorithms, parallel algorithms with communication methods often have faster convergence speeds and more accurate fitness values than native algorithms. Parallel processing is a very important way to deal with computer optimization. This is a form of calculation that runs simultaneously in the same direction. To construct a parallel structure, this paper divides the population into iterative ways, divides the entire population into several sub-populations according to a predetermined method, iterates and updates continuously according to the set target fitness value, and generates the next generation from the result. The parallel processing results proposed in this paper require constant exchange of attributes between their sub-populations, such as operations, substitution, exchange, movement, or mutation Algorithm 1 is a communication strategy proposed in this paper. The excellent solution and the poor solution in the solution space are operated, and the search is continuously performed in the search space.

---

**Algorithm 1** A pseudo-code of a parallel with communication strategies.

1: **if** $m > 2$ **then**//$m$ is the number of pigeons in each group, and c is 0.5.
2:     **if** *random* $< c$ **then**//Stategy1- neighboring groups
3:         **for** $i = 1$ to $m$ **do**
4:             Replace the worst ($G_i$) with the best in each group ($G_j, j \neq i$)
5:         **end for**
6:     **end if**
7:     **if** *random* $>= c$ **then**//Stategy2- the best to all
8:         **for** $i = 1$ to $m$ **do**
9:             Use the best variation in each group the worst in each group
10:         **end for**
11:     **end if**
12: **end if**

---

All communication strategies are substituted for all sub-populations in the population, and the best and worst among the sub-populations are updated. In the replacement process, the best pigeons of all sub-populations will migrate to each group, mutate them by replacing the worst pigeons in each group, and update them after a specified period. The strategy with neighbouring populations is to migrate the best pigeons in each population to adjacent populations, and then replace the poorer pigeons in adjacent populations.

### 4.2. Compacted Pigeon-Inspired Optimization

In the estimated distribution algorithm (EDA), the probabilistic representation is used to obtain fewer storage variables, and the overall population of all solutions is stored in the PIO algorithm, while still obtaining the same optimization results. Compact uses the above-mentioned EDA principle to simulate the iterative behavior of the population for meta-heuristic calculations. The probabilistic model, to represent the population-based algorithms, operate in a compact model. In this case, in the compact model algorithm, the actual population is treated as a virtual population. Virtual populations are set based on EDA's probability density functions (PDFs). Not all solutions are stored in memory, but some new candidate solutions are generated based on the probability distribution of the solutions in memory. When a new candidate solution is generated, it is generated in a solution space with a relatively good fitness value. In the compact algorithm, the possibility of individuals and virtual populations indicates that learning from the previous generation will then affect the next generation of solutions. In this paper, the vector of the generated solution is called Perturbation Vector (PV). These principles are applied to the improvement of compact PIO memory variables.

Different from the original PIO, the compact method [49–53] uses the encoded data structure of the probability vector and calls the population a virtual population. New candidate solutions are generated in the process of constantly changing probability, which affects the change of probability. The compact algorithm's optimization processing goal is to simulate the behavior of pigeons, but its storage unit variable memory is much smaller than the original algorithm. PV continuously generates candidate solutions from vector probabilities. The updated probability vector will reflect the changing process of the solution. PV is a matrix that includes two specified parameters, average $\mu$ and quasi deviation $\sigma$, in the *PDF*. $t$ is the number of iterations, $\mu$ and $\sigma$ values are in the range of $[-1, 1]$, and it can be defined as: $PV^t = [\mu^t, \sigma^t]$.

$$PDF = \frac{e^{-\frac{(x-\mu[k])^2}{2*\sigma[k]^2}} * \sqrt{\frac{2}{\pi}}}{\sigma[k] * (erf(\frac{\mu[k]+1}{\sqrt{2}*\sigma[k]}) - erf(\frac{\mu[k]-1}{\sqrt{2}*\sigma[k]})))} \tag{18}$$

where *PDF* is the probability distribution function of *PV*, $\mu$ and $\sigma$ are two parameters of Equation (18), which control the probability change of *PDF*. The definition of *erf* can be found in [37]. *PDF* in this paper is achieved by constructing a polynomial cumulative distribution function (CDF) [54].

### 4.3. Parallel Compact Pigeon-Inspired Optimization

This section introduces the implementation of hybrid parallel and compact PIO algorithms. Constructing a parallel method, the entire pigeon population can be divided into several sub-populations in a predetermined manner, and there is a point to stimulate when communicating between the sub-populations. This method is based on the optimization of pigeon populations, and the sub-populations are run in parallel. The independent evolutions of pigeons do not affect each other. When performing inter-group communication, for example, when the best pigeons in one sub-population are moved to another sub-population, they can be replaced with the weakest pigeons based on the measured fitness, and the subgroups are updated during this period. In the compact phase, we calculate the agreement on the sub-population based on the probability vector through competition.

New candidate solutions are generated based on the distribution of the previous generation of the *PDF* samples. It can be said that the PV operation will randomly generate the position of the pigeon.

Algorithm 2 can represent the initialization PV operation as the CPIO pseudo code, and train the probability generation $Pos_{gbest}$ based on the measured results. In the algorithm, $c$ is a relatively large constant, and it is often set to 10.

---
**Algorithm 2** Initialization of CPIO.

---
1: Initialization of $PV(\mu, \sigma)$
2: **for** $i = 1$ to $D_{im}$ **do**
3: 　　$\mu_i^t = 0$;
4: 　　$\sigma_i^t = c$;
5: **end for**
6: Initializing pigeons location Pos via PV
7: Initializing $Pos_{gbest}$ with the best location value: $Pos_{gbest}$ = arg minf[x].

---

The *Winner* and *Loser* variables are constantly changing. When a new solution is generated, it will be compared with the best global solution that has been generated. The better solution will become the new *Winner*, and the worse solution will become the new *Loser*. The evaluation criterion is based on the degree of fitness. Based on the above comparison,*PDF* will transfer the winner to a region in the search space that is more promising to produce a better solution. The newly selected candidate object is evaluated by the value of the fitness value. Algorithm 3 shows what is described in this paragraph.

---
**Algorithm 3** Compete for winner and loser.

---
1: **if** fitness $(Pos_{gbest}^t)$ > fitness $(Pos^{t+1})$ **then**
2: 　　*Winner* is set to $Pos^{t+1}$
3: 　　*Loser* is set to $Pos_{gbest}^t$
4: **else**
5: 　　*Winner* is set to $Pos_{gbest}^t$
6: 　　*Loser* is set to $Pos^{t+1}$
7: **end if**

---

In addition, the two vectors in the PV operation will be continuously updated according to the changes of *Winner* and *Loser* in Equations (19) and (20). A parameter called a virtual population does not correspond to a strict variable of a population size variable such as in a population-based algorithm.

$$\mu_i^{t+1} = \mu_i^t + \frac{winner_i - loser_i}{N} \tag{19}$$

$$\sigma_i^{t+1} = \sqrt{(\sigma_i^t)^2 + (\mu^{t+1})^2 + \frac{winner_i^2 - loser_i^2}{N}} \tag{20}$$

Compare with $Pos_{gbest}$, and use the selected position *Pos* to evaluate the fitness function to obtain the next-generation winner solution. The update operation of PV is shown in Algorithm 4.

---
**Algorithm 4** Updating PV for new candidates.

---
1: **for** $i = 1$ to $D_{im}$ **do**
2: 　　$\mu_b = \mu$
3: 　　$\mu_i^{t+1} = \mu_i^t + \frac{winner_i - loser_i}{N}$
4: 　　$\sigma_i^{t+1} = \sqrt{(\sigma_i^t)^2 + (\mu_b^{t+1})^2 + \frac{winner_i^2 - loser_i^2}{N}}$
5: **end for**

---

Algorithm 5 represents the pseudo code of the steps of the compact PIO. Sampling by probability model simulates the population-based iterative process of the original PIO. The virtual population is represented in its probability.

---

**Algorithm 5** The Compact PIO (CPIO).

---

  1:  Initialization phase according to Algorithm 2
  2:  **while** stop criteria are not met **do**
  3:      Generating *Pos* by *PV*
  4:      Enter map and compass operator;
  5:      Update Pigeons via Equations (11) and (12)
  6:      Select $Pos_{gbest}$ by Compete via Algorithm 3
  7:      [winner, loser] = compete (*Pos*, *newPos*);
  8:      Fnew=f (newPos);
  9:      Update PV scheme $\mu^{t+1}$, $\sigma^{t+1}$, via Algorithm 4
10:      Global best update;
11:      [winner, loser] = complete (*newPos*, $Pos_{gbest}$);
12:      Update *winner* and *loser*;
13:      Enter landmark operator
14:      Finding virtual pigeons with the ability to know the way according to the iterative process of the first stage
15:      Virtual population shrinking via Equation (14)
16:      Update Pigeons via Equation (15)
17:      Select $Pos_{gbest}$ by Compete via Algorithm 3
18:      [winner, loser] = compete (*Pos*, *newPos*);
19:      Fnew=f (newPos);
20:      Update PV scheme $\mu^{t+1}$, $\sigma^{t+1}$, via Algorithm 4
21:      Global best update;
22:      [winner, loser] = complete (*newPos*, $Pos_{gbest}$);
23:      Update *winner* and *loser*;
24:      t = t + 1;
25:  **end while**

---

The simplified steps for Hybrid Parallel and compact PIO (PCPIO) are as follows. First, the entire population is divided into G sub-populations, and the objective function fitness of the algorithm and the communication cycle of the algorithm are set. Next, enter the compact policy and assess the results of the current to find the best solution. Finally, check whether the current number of iterations meets the setting. If it is not satisfied, go to the second step to continue the iteration; otherwise, the position of the best pigeon and the output fitness value will be recorded. Algorithm 6 shows the overall pseudo code of PCPIO, where *G* is the number of sub-populations, $N_p$ is the number of pigeons per group, *R* is the map and compass operator, usually set to 0.2, *T* is the inter-species communication cycle, and *CPIO* is the compact program.

---

**Algorithm 6** Pseudo code for Parallel and Compact Pigeon-Inspired Optimization (*PCPIO*).

---

  1:  Initialization
  2:  generate $G_{1...m}$ $m <= N_p$ subgroups, each *G* has $n = N_p/G$ pigeons
  3:  set exchanging time T, counter=1;
  4:  solutions $Pos_{i,j}^t$ in the j-th sub-group with *m* pigeons, i = 1, 2, ..., *m*; j = 1, 2, ..., *n*
  5:  **while** termination is not satisfied **do**
  6:      **for** $i = 1$ to *m* **do**
  7:         *CPIO* according to Algorithm 5
  8:      **end for**
  9:      **if** mod (t,T)==0 **then**
10:         *communication* according to Algorithm 1
11:         Find the current best fitness solution $Pos_{gbest}$
12:      **end if**
13:      t+1
14:  **end while**

---

## 5. Experiment with Numerical Optimization Problems

To evaluate the performance of the PCPIO algorithm proposed in this paper, the test function in [19,37] was used as the criterion. Detailed information about the function is given in Tables 1–3, where **TM** represents the theoretical minimum of the test function. PCPIO, PIO, PSO, and CPIO were compared. Table 4 gives detailed experimental results. The number of iterations of PIO and its improved algorithm were 120 times in the first stage, 80 times in the second stage, and 200 times of PSO operation.

**Table 1.** Single peak in test function.

| Number | Function | Space | Dimension | TM |
|:---:|:---:|:---:|:---:|:---:|
| 1 | $F_1(y) = \sum_{j=1}^{No} y_j^2$ | $[-100, 100]$ | 30 | 0 |
| 2 | $F_2(y) = \sum_{j=1}^{No} \left|y_j\right| + \prod_{j=1}^{No} \left|y_j\right|$ | $[-10, 10]$ | 30 | 0 |
| 3 | $F_3(y) = \sum_{j=1}^{No} \left(\sum_{k=1}^{j} y_k\right)^2$ | $[-100, 100]$ | 30 | 0 |
| 4 | $F_4(y) = max_j \left|y_j\right|, j \in [1, m]$ | $[-100, 100]$ | 30 | 0 |
| 5 | $f_5(y) = \sum_{j=1}^{No-1} \left[100 \left(y_{j+1} - y_j^2\right)^2 + \left(y_j - 1\right)^2\right]$ | $[-30, 30]$ | 30 | 0 |
| 6 | $f_6(y) = \sum_{j=1}^{No} \left(\left[y_j + 0.5\right]\right)^2$ | $[-100, 100]$ | 30 | 0 |
| 7 | $f_7(y) = \sum_{j=1}^{No} j * y_j^2 + rand[0, 1)$ | $[-1.28, 1.28]$ | 30 | 0 |

**Table 2.** Multimodality peak in test function.

| Number | Function | Space | Dimension | TM |
|:---:|:---:|:---:|:---:|:---:|
| 8 | $F_8(y) = \sum_{j=1}^{No} -y_j * sin\left(\sqrt{\left|y_j\right|}\right)$ | $[-500, 500]$ | 30 | $-12{,}569$ |
| 9 | $f_9(y) = \sum_{j=1}^{No} \left[y_j^2 - 10 * cos\left(2\pi y_j\right) + 10\right]$ | $[-5.12, 5.12]$ | 30 | 0 |
| 10 | $f_{10}(y) = -20 * exp\left(-0.2\sqrt{\frac{1}{No}\sum_{j=1}^{No} y_j^2}\right)$ $-exp\left(\frac{1}{No}\sum_{j=1}^{No} cos\left(2\pi y_j\right) + 20 + 2.718\right)$ | $[-32, 32]$ | 30 | 0 |
| 11 | $f_{11}(y) = \frac{1}{4000} * \sum_{j=1}^{No} y_j^2 - \prod_{j=1}^{No} cos\left(\frac{y_j}{\sqrt{j}}\right) + 1$ | $[-600, 600]$ | 30 | 0 |
| 12 | $f_{12}(y) = \frac{\pi}{No} * \left\{10 * sin\left(\pi y_1\right) + \right.$ $\sum_{j=1}^{No-1} \left(y_j - 1\right)^2 \left[1 + 10 * sin^2(\pi y_{j+1})\right] + \left.(y_{No} - 1)^2\right\}$ $+ \sum_{j=1}^{No} u(y_j, 10, 100, 4),$ $y_j = 1 + \frac{y_j+1}{4} * u(z_l, a, k, m) = \begin{cases} k(y_j - a), y > a \\ 0, -a < y_j < a \\ k(-y_j - a), y > a \end{cases}$ | $[-50, 50]$ | 30 | 0 |
| 13 | $f_{13}(y) = 0.1 *$ $\left\{sin^2\left(3\pi y_1\right) + \sum_{j=1}^{No} \left(y_j - 1\right)^2 \left[1 + sin^2\left(3\pi y_j + 1\right)\right]\right.$ $+ (y_{No} - 1)^2 \left[1 + sin^2\left(2\pi y_{No}\right)\right]\Big\}$ $+ \sum_{j=1}^{No} u\left(y_j, 10, 100, 4\right)$ | $[-50, 50]$ | 30 | 0 |

**Table 3.** Others peak in test function.

| Number | Function | Space | Dimension | TM |
|--------|----------|-------|-----------|-----|
| 14 | $f_{14}(y) = \left( \frac{1}{500} * \sum_{j=1}^{25} \frac{1}{j + \sum_{k=1}^{2} \left( z_k - a_{kj} \right)^6} \right)^{-1}$ | $[-65, 65]$ | 2 | 1 |
| 15 | $f_{15}(y) = \sum_{j=1}^{11} \left[ a_j - \frac{y_1 \left( b_j^2 + b_j y^2 \right)}{b_j^2 + b_j y^3 + y^4} \right]^2$ | $[-5, 5]$ | 4 | 0.00030 |
| 16 | $f_{16}(y) = 4y_j^2 - 2.1y_j^4 + \frac{1}{3}y_j^6$ $+ y_j y_2 - 4y_2^2 + 4y_2^4$ | $[-5, 5]$ | 2 | $-1.0316$ |
| 17 | $f_{17}(y) = \left( y_2 - \frac{5.1}{4\pi^2} y_j^2 + \frac{5}{\pi} y_j - 6 \right)^2$ $+ 10 \left( 1 - \frac{1}{8\pi} \right) \cos y_j + 10$ | $[-5, 5]$ | 2 | 0.398 |
| 18 | $f_{18}(y) = \left[ 1 + (y_1 + y_2 + j)^2 * \right.$ $\left( 19 - 14y_1 + 3y_1^2 - 14y_2 + 6y_1 y_2 + 3y_2^2 \right) \right]$ $* \left[ 30 + (2y_1 - 3y_2)^2 * \right.$ $\left. \left( 18 - 32y_1 + 12y_1^2 + 48y_2 - 36y_1 y_2 + 27y_2^2 \right) \right]$ | $[-2, 2]$ | 2 | 3 |
| 19 | $f_{19}(y) = -\sum_{j=1}^{4} c_j * exp \left( -\sum_{k=1}^{3} a_{jk} \left( y_k - p_{jk} \right)^2 \right)$ | $[1, 3]$ | 3 | $-3.86$ |
| 20 | $f_{20}(y) = -\sum_{j=1}^{4} c_j * exp \left( -\sum_{k=1}^{6} a_{jk} \left( y_k - p_{jk} \right)^2 \right)$ | $[0, 1]$ | 6 | $-3.32$ |
| 22 | $f_{22}(y) = -\sum_{j=1}^{7} \left[ \left( y - a_j \right) \left( y - a_j \right)^T + c_j \right]^{-1}$ | $[0, 10]$ | 4 | $-10.4028$ |
| 23 | $f_{23}(y) = -\sum_{j=1}^{10} \left[ \left( y - a_j \right) \left( y - a_j \right)^T + c_j \right]^{-1}$ | $[0, 10]$ | 4 | $-10.5363$ |
| 21 | $f_{21}(y) = -\sum_{j=1}^{5} \left[ \left( y - a_j \right) \left( y - a_j \right)^T + c_j \right]^{-1}$ | $[0, 10]$ | 4 | $-10.1532$ |

**Table 4.** Comparison of OPIO, PPIO, PC-PIO, and PSO algorithms.

| Test Function | OPIO | | CPIO | | PCPIO | | PSO | |
|------|------|-----|------|-----|-------|-----|-----|-----|
| | Mean | Std | Mean | Std | Mean | Std | Mean | Std |
| F1 | $4.31 \times 10^2$ | $1.29 \times 10^2$ | $4.14 \times 10^{-1}$ | $8.14 \times 10^{-2}$ | $4.92 \times 10^{-3}$ | $4.02 \times 10^{-3}$ | $1.02 \times 10^3$ | $6.12 \times 10^4$ |
| F2 | $9.12 \times 10^{-1}$ | $2.08 \times 10^0$ | $2.80 \times 10^0$ | $3.55 \times 10^{-1}$ | $3.33 \times 10^{-1}$ | $1.72 \times 10^{-1}$ | $1.45 \times 10^1$ | $1.17 \times 10^1$ |
| F3 | $2.97 \times 10^3$ | $3.45 \times 10^3$ | $1.18 \times 10^0$ | $3.79 \times 10^{-1}$ | $5.40 \times 10^{-2}$ | $3.12 \times 10^{-2}$ | $4.88 \times 10^3$ | $3.75 \times 10^6$ |
| F4 | $4.04 \times 10^{-1}$ | $4.77 \times 10^{-1}$ | $2.89 \times 10^1$ | $3.56 \times 10^{-2}$ | $3.24 \times 10^{-2}$ | $2.30 \times 10^{-2}$ | $1.65 \times 10^1$ | $5.98 \times 10^0$ |
| F5 | $1.50 \times 10^4$ | $4.56 \times 10^4$ | $6.62 \times 10^1$ | $9.29 \times 10^0$ | $2.95 \times 10^1$ | $3.02 \times 10^{-1}$ | $9.01 \times 10^4$ | $5.34 \times 10^9$ |
| F6 | $3.30 \times 10^1$ | $9.31 \times 10^1$ | $5.00 \times 10^0$ | $5.72 \times 10^{-1}$ | $1.95 \times 10^0$ | $2.01 \times 10^{-1}$ | $1.00 \times 10^3$ | $1.07 \times 10^5$ |
| F7 | $4.48 \times 10^{-3}$ | $1.31 \times 10^{-2}$ | $8.10 \times 10^{-1}$ | $3.06 \times 10^{-1}$ | $1.50 \times 10^{-3}$ | $1.81 \times 10^{-3}$ | $1.51 \times 10^{-1}$ | $1.22 \times 10^{-2}$ |
| F8 | $-4.08 \times 10^3$ | $6.21 \times 10^2$ | $-7.18 \times 10^3$ | $1.59 \times 10^3$ | $-8.56 \times 10^3$ | $3.52 \times 10^3$ | $-5.67 \times 10^3$ | $5.05 \times 10^5$ |
| F9 | $1.33 \times 10^1$ | $2.79 \times 10^1$ | $1.53 \times 10^2$ | $5.68 \times 10^1$ | $2.14 \times 10^{-5}$ | $7.85 \times 10^{-6}$ | $5.89 \times 10^1$ | $2.33 \times 10^2$ |
| F10 | $8.42 \times 10^{-1}$ | $1.60 \times 10^0$ | $1.06 \times 10^0$ | $1.76 \times 10^{-1}$ | $6.29 \times 10^{-2}$ | $2.76 \times 10^{-2}$ | $8.18 \times 10^0$ | $8.20 \times 10^{-1}$ |
| F11 | $5.67 \times 10^{-1}$ | $4.21 \times 10^{-1}$ | $1.93 \times 10^{-4}$ | $4.54 \times 10^{-4}$ | $5.16 \times 10^{-4}$ | $3.63 \times 10^{-4}$ | $9.45 \times 10^0$ | $8.90 \times 10^0$ |
| F12 | $1.00 \times 10^0$ | $1.06 \times 10^0$ | $1.18 \times 10^{-1}$ | $8.88 \times 10^{-2}$ | $6.12 \times 10^{-1}$ | $6.27 \times 10^{-1}$ | $6.44 \times 10^0$ | $5.47 \times 10^0$ |
| F13 | $2.36 \times 10^0$ | $2.29 \times 10^0$ | $2.33 \times 10^{-1}$ | $1.02 \times 10^{-1}$ | $1.14 \times 10^0$ | $1.24 \times 10^0$ | $3.85 \times 10^0$ | $1.62 \times 10^8$ |
| F14 | $4.37 \times 10^0$ | $3.28 \times 10^0$ | $1.27 \times 10^1$ | $1.24 \times 10^{-9}$ | $8.15 \times 10^0$ | $4.09 \times 10^0$ | $2.55 \times 10^0$ | $3.73 \times 10^0$ |
| F15 | $1.57 \times 10^{-3}$ | $1.40 \times 10^{-3}$ | $3.73 \times 10^0$ | $2.17 \times 10^{-3}$ | $9.10 \times 10^{-3}$ | $9.76 \times 10^{-3}$ | $1.10 \times 10^{-3}$ | $4.00 \times 10^{-6}$ |
| F16 | $-1.03 \times 10^0$ | $2.93 \times 10^{-12}$ | $-1.02 \times 10^0$ | $1.13 \times 10^{-2}$ | $-1.03 \times 10^0$ | $4.90 \times 10^{-4}$ | $-1.03 \times 10^0$ | $4.17 \times 10^{-31}$ |
| F17 | $4.04 \times 10^{-1}$ | $4.10 \times 10^{-2}$ | $5.62 \times 10^{-1}$ | $1.44 \times 10^1$ | $3.98 \times 10^{-1}$ | $1.87 \times 10^0$ | $3.98 \times 10^{-1}$ | $2.59 \times 10^{-2}$ |
| F18 | $3.00 \times 10^0$ | $7.83 \times 10^{-3}$ | $8.74 \times 10^0$ | $4.02 \times 10^0$ | $3.74 \times 10^0$ | $3.15 \times 10^0$ | $3.00 \times 10^0$ | $2.65 \times 10^{-30}$ |
| F19 | $-3.80 \times 10^0$ | $2.92 \times 10^{-2}$ | $-3.54 \times 10^0$ | $4.68 \times 10^{-1}$ | $-3.80 \times 10^0$ | $6.91 \times 10^{-2}$ | $-3.81 \times 10^0$ | $2.29 \times 10^{-3}$ |
| F20 | $-2.29 \times 10^0$ | $4.39 \times 10^{-1}$ | $-2.38 \times 10^0$ | $3.20 \times 10^{-1}$ | $-2.97 \times 10^0$ | $2.76 \times 10^{-1}$ | $-2.09 \times 10^0$ | $1.81 \times 10^{-1}$ |
| F21 | $-5.48 \times 10^0$ | $2.07 \times 10^0$ | $-2.62 \times 10^0$ | $8.83 \times 10^{-1}$ | $-5.04 \times 10^0$ | $1.07 \times 10^{-2}$ | $-1.62 \times 10^0$ | $1.07 \times 10^0$ |
| F22 | $-5.29 \times 10^0$ | $1.78 \times 10^0$ | $-2.56 \times 10^0$ | $9.26 \times 10^{-1}$ | $-5.08 \times 10^0$ | $6.23 \times 10^{-3}$ | $-1.54 \times 10^0$ | $3.48 \times 10^{-1}$ |
| F23 | $-4.87 \times 10^0$ | $1.66 \times 10^0$ | $-2.48 \times 10^0$ | $8.89 \times 10^{-1}$ | $-5.12 \times 10^0$ | $8.45 \times 10^{-3}$ | $-1.92 \times 10^0$ | $5.85 \times 10^{-1}$ |

The simulation method proves that the method works well, but the lack of real photovoltaic equipment to perform the experiment will have more parameters affecting the experimental results. The experiments were performed on a system with Windows 10 and a CPU of i7-4710MQ 2.5 GHZ. The software environment was MATLAB R2016b.

The parameters of the algorithm were set as follows. The virtual population size was 120. The number of dimensions in the solution space was set according to the references. Each test function was required to run 30 times, and its average and variance were taken after the end of the run. The entire population communicates once every 20 times. PIO, CPIO, PCPIO, and PSO were compared with experiments. From the comparison results in Figures 4–6, PCPIO has faster convergence speed and more accurate target values in high dimensions. Comparing PCPIO with other algorithms in this paper, for the convergence speed and the optimal value reached, more than half of the test functions analyzed from the experimental results show that PCPIO is quite competitive. PCPIO has greater competitiveness.

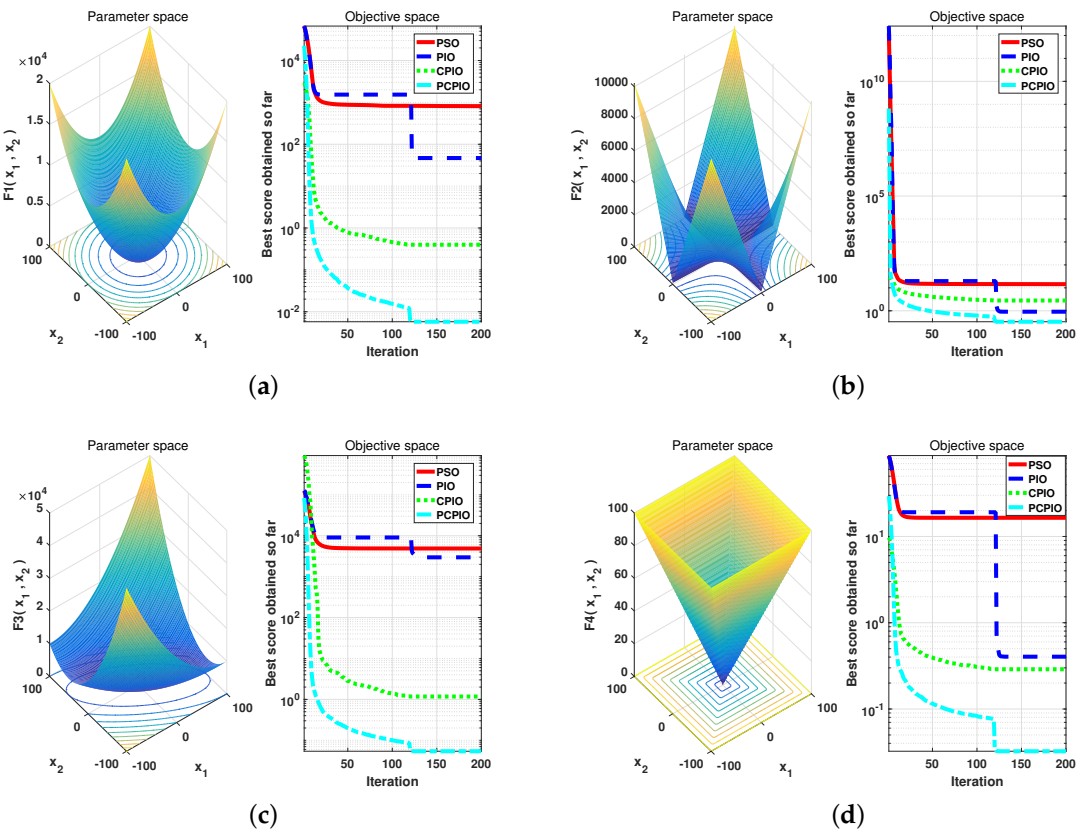

**Figure 4.** Comparison of running times of the PCPIO, with the CPIO, PIO, and PSO algorithms in the test functions. (**a**): f1 function, (**b**): f2 function, (**c**): f3 function, (**d**): f4 function.

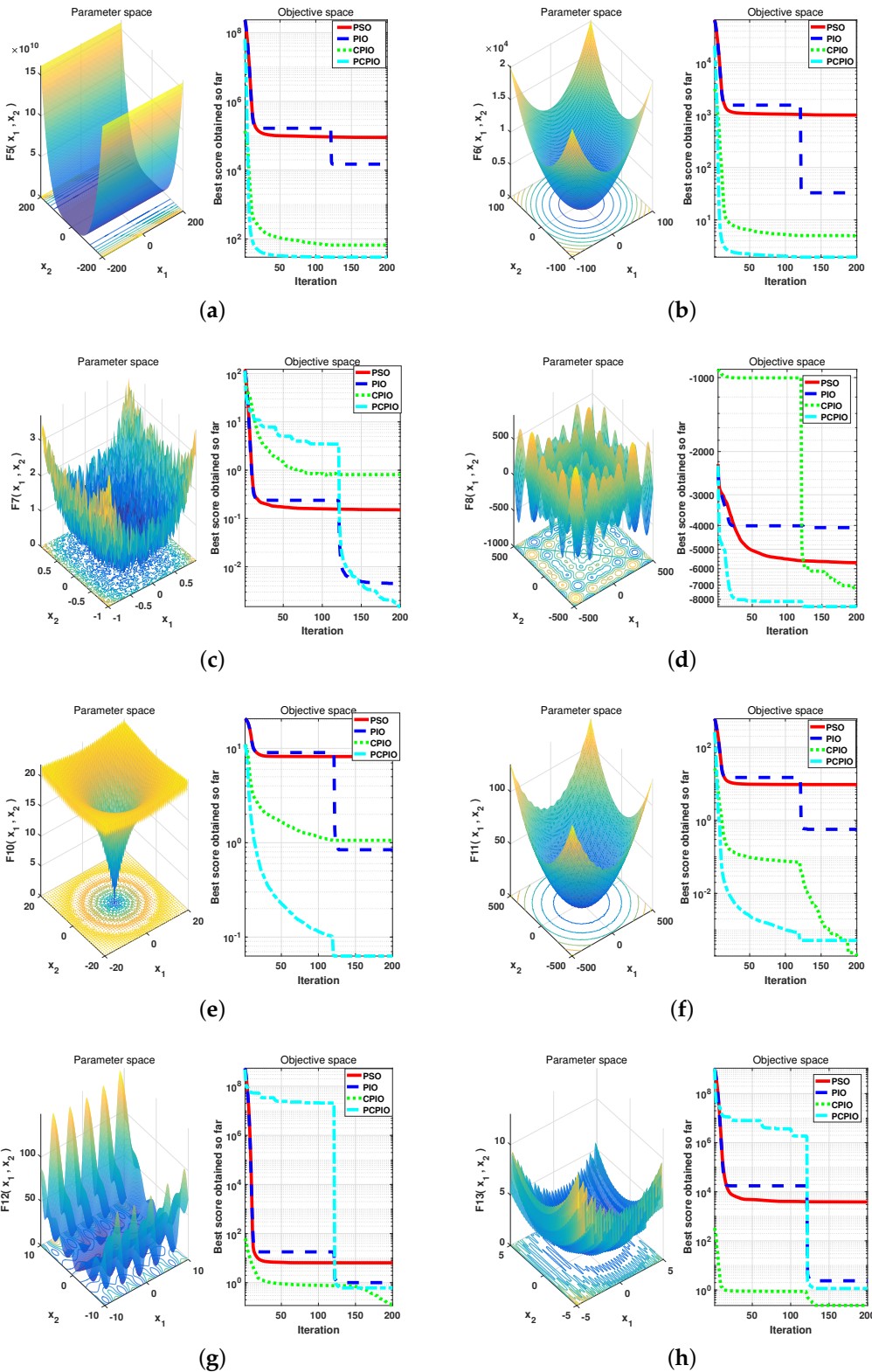

**Figure 5.** Comparison of running times of the PCPIO, with the CPIO, PIO, and PSO algorithms in the test functions. (**a**): f5 function, (**b**): f6 function, (**c**): f7 function, (**d**): f8 function, (**e**): f10 function, (**f**): f11 function, (**g**): f12 function, (**h**): f13 function.

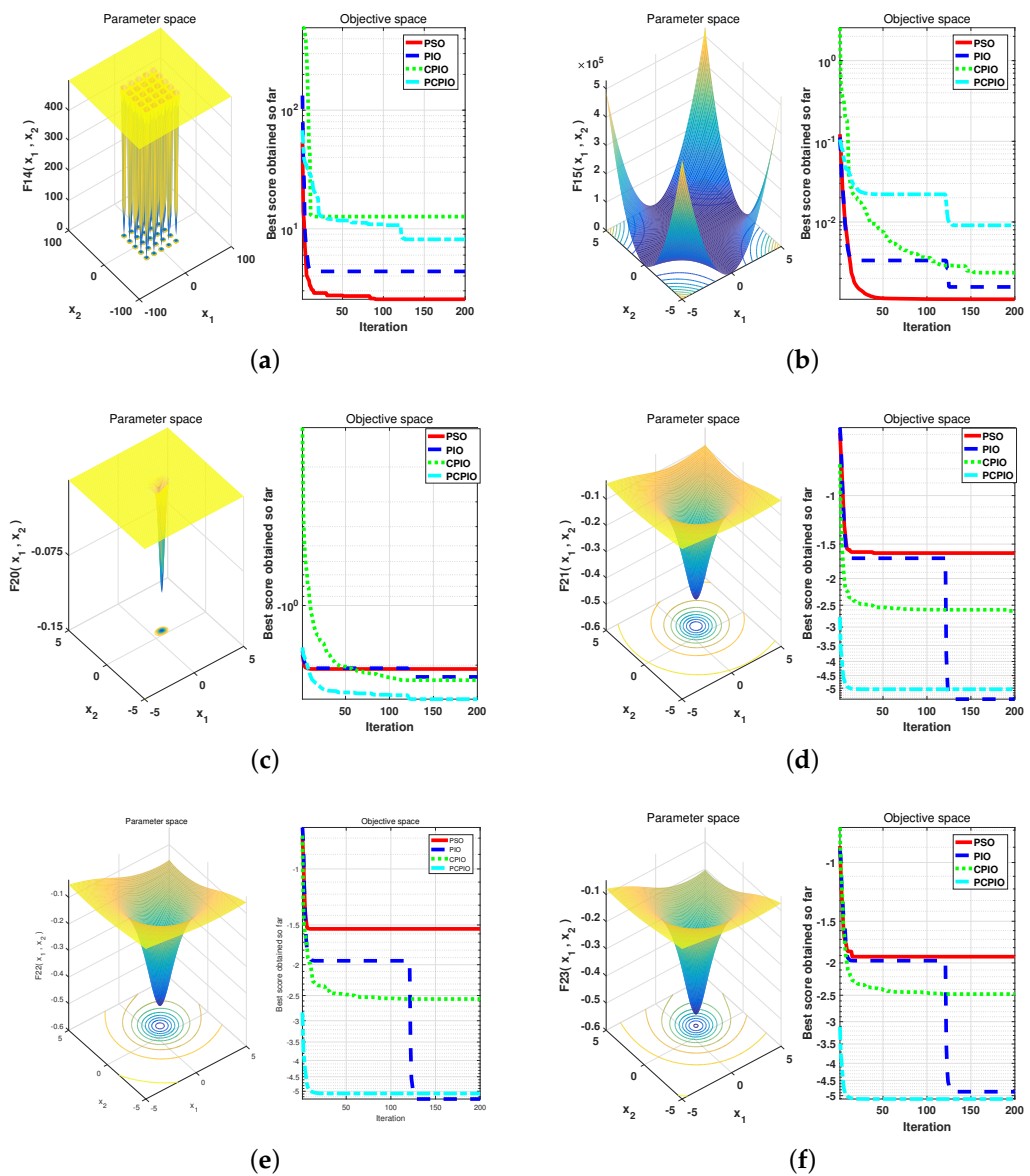

**Figure 6.** Comparison of running times of the PCPIO, with the CPIO, PIO, and PSO algorithms in the test functions. (**a**): f14 function, (**b**): f15 function, (**c**): f20 function, (**d**): f21 function, (**e**): f22 function, (**f**): f23 function.

## 6. Applied PCPIO Based MPPT Technique for PV Systems

*Applied for PV Systems*

As a new type of bionic evolutionary algorithm, the working process of the PIO algorithm is to iteratively update the parameters and finally approach the optimal solution. In addition, while the PIO algorithm optimizes global parameters, it can also identify each pigeon as a solution to solve the optimization and search problems. The solution to this problem is the result of elimination after each pigeon evolved and compared. Under complex lighting conditions, the PU characteristic curve of the photovoltaic array becomes a multi-peak state. The traditional single-dimensional algorithm cannot distinguish the difference between the extreme values, which may lead to local optimization and cause power loss. Figures 7 and 8 show the characteristics of P–V and I–V with cover and MPP optimization using PSO and PCPIO. The results show that PCPIO is better than PSO to find the maximum power point that PCPIO and PSO can reach with cover. The basic principle of PCPIO-based MPPT can be

explained as: during the iterative operation of the objective function, each pigeon records its own speed and compares with its previous position, and finds the closest position to the MPP, which is the optimal value of the group of pigeons *gbest*. After the elimination and evolution of the particles, when *gbest* reaches the MPP, the function iteration terminates.

$$P_{PV} = \sum_{k=1}^{N} V_{pv_k} I_{pv_k} - \hat{r}_{s_k} i_{L_k}^2 - \frac{V_{pv_k}^2}{\hat{R}_{p_k}} \tag{21}$$

Equation (21) is the fitness function, while parameters $\hat{r}_{s_k}$ and $\hat{R}_{p_k}$ are the estimated equivalent losses parameters.

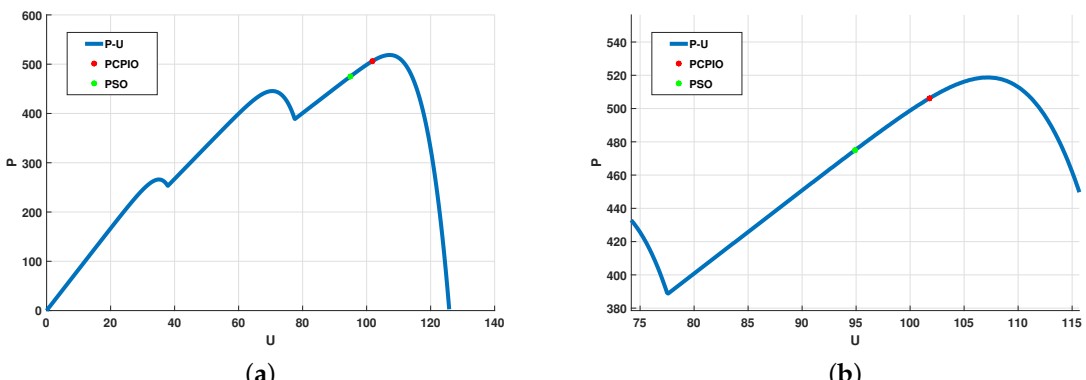

**Figure 7.** P–V curve of PV array under partial shadow and standard environment. (**a**): PV array under standard environment (**b**): PV array under shadow environment

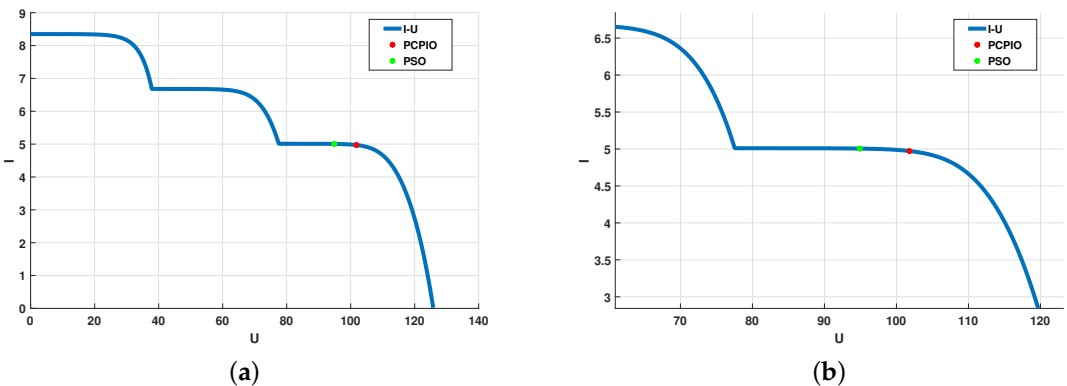

**Figure 8.** I–V curve of PV array under partial shadow and standard environment. (**a**): PV array under standard environment (**b**): PV array under shadow environment

PCPIO performs better under complex lighting conditions, and has the following specific performances: (1) PCPIO algorithm has the ability of global optimization when comparing single-dimensional MPPT algorithm. (2) When comparing PCPIO algorithms with multi-dimensional intelligent algorithms, there are not many parameters, the equations are simple and easy to understand, and it is easy to implement in engineering. (3) In the PCPIO algorithm, setting the inertia weight and the number of particles can control the convergence speed, each particle has memory, and the solution spaces do not interfere with each other.

To conveniently analyze and study the volt–ampere characteristics and external characteristics of photovoltaic cells, we must first model the photovoltaic cells, including photovoltaic cell size, system maximum voltage value, peak power, maximum power range, open circuit voltage, short circuit

current, and current and voltage values at maximum power. Using these parameters and the above equation model, a photovoltaic cell simulation model was built in MATLAB.

Figure 9 shows the MPPT algorithm model is shown. Figure 10 shows the PCPIO-based MPPT algorithm model. The specific model parameters are shown in Table 5.

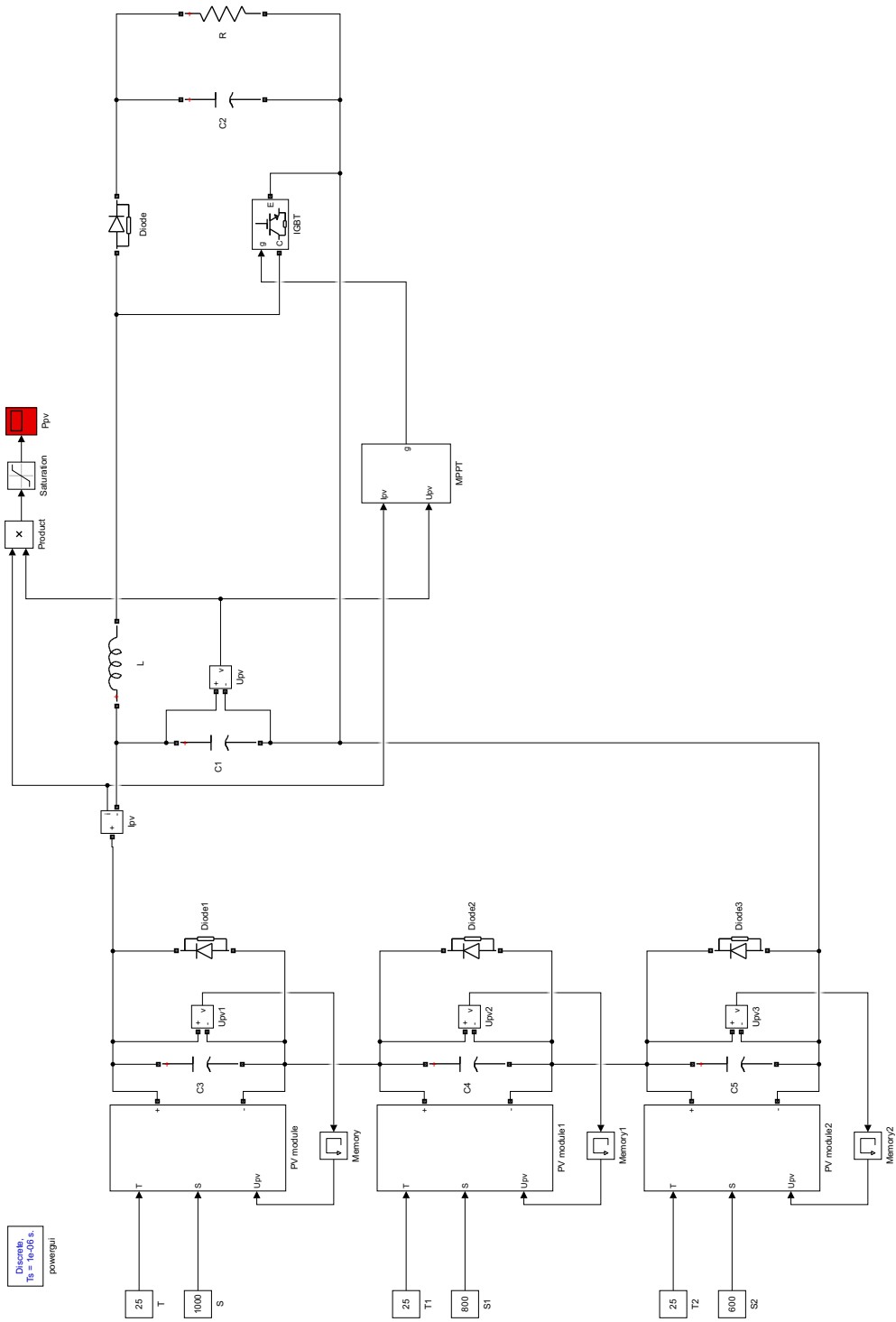

**Figure 9.** The part of the simulation model of MPPT algorithm.

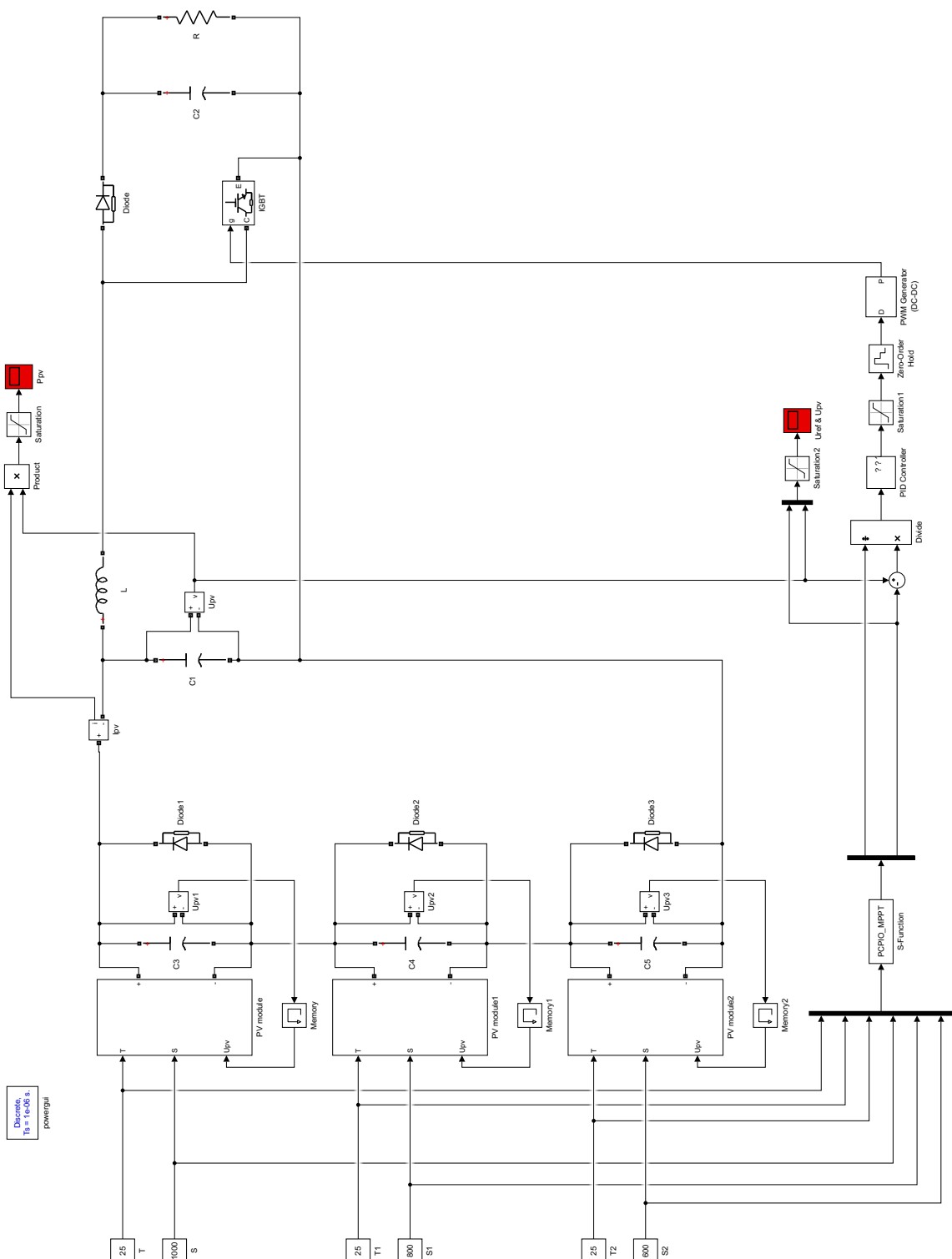

**Figure 10.** The part of the simulation model of PCPIO-based MPPT algorithm.

**Table 5.** The chart of PV array's parameter settings.

| Parameter | Value |
|---|---|
| Open circuit voltage $U_{oc}$ | 43.6 V |
| Short-circuit current $I_{sc}$ | 8.35 A |
| Maximum power point voltage $U_{mp}$ | 35 V |
| Maximum power point current $I_{mp}$ | 7.6 A |

At the installation site of the photovoltaic array, the temperature and light will inevitably change at any time. Moreover, the shadows generated by moving clouds and other parts will partially block the photovoltaic array and other similar complex lighting conditions. This situation leads to a multi-peak state of the PV characteristic curve of the photovoltaic array, which not only reduces the power generation efficiency of the photovoltaic array, but also the power oscillations it generates and it will reduce the life of the related photovoltaic elements. To solve this problem, this paper proposes PCPIO and MPPT hybrid algorithm.

MPPT was performed on photovoltaic arrays using ordinary perturbation observation methods. The array was composed of three above-mentioned modules in series. The three photovoltaic modules had a temperature of 25 °C and a light intensity of 1000, 800, and 600 W/m$^2$. The simulation model uses an algorithm that directly perturbs the duty cycle. As can be seen in Figure 11, the power of the pure MPPT algorithm is finally stable at about $1.25 \times 10^2$ W, while the hybrid algorithm of PCPIO and MPPT is stable at about $5.15 \times 10^2$ W. Although the traditional variable-step-disturbance observation method is fast and stable, it does not have the ability of global optimization, and the PCPIO algorithm can make up for the above shortcomings.

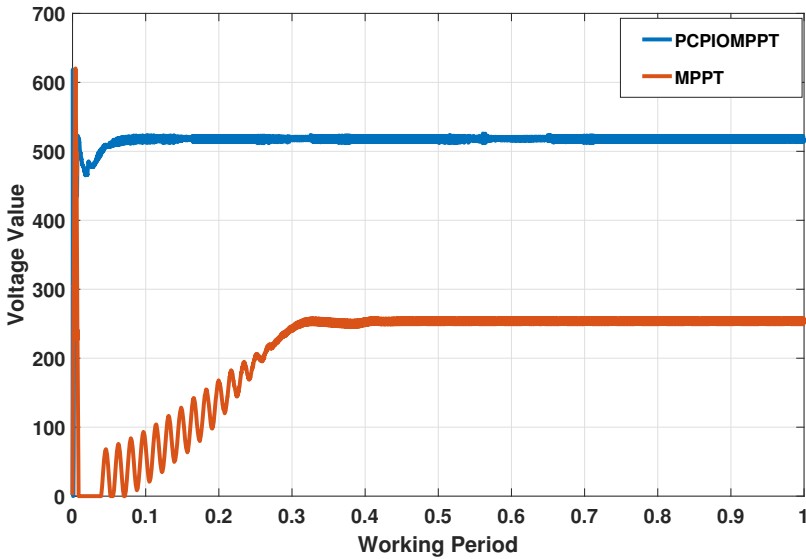

**Figure 11.** Maximum power curve of photovoltaic power generation under the control of MPPT algorithm and PCPIO and MPPT hybrid algorithm.

## 7. Conclusions

This paper evaluates and improves the global MPPT performance based on the PCPIO algorithm. The proposed algorithm aims to increase the maximum power that can be achieved by photovoltaic power generation under partially shaded conditions. The proposed photovoltaic system is composed

of MPPT system, boost converter, and photovoltaic array, and is implemented in Matlab/Simulink software. The tracking efficiency and overall performance of the MPPT technology studied were evaluated. According to the experimental results, compared with the traditional algorithm, the tracking algorithm based on PCPIO has high accuracy and stability in extracting global MPP. On the other hand, this paper improves the PIO algorithm, a new hybrid parallel and compact PIO algorithm, and applies it to the maximum power point tracking of photovoltaic power generation. The new algorithm performs very well in the test function. This method avoids most local solution problems in compound constrained optimization problems, and has faster convergence speed and more accurate accuracy. The experimental results provide a new method for tracking the maximum power point of photovoltaic power generation.

**Author Contributions:** Conceptualization, J.-S.P. and Y.L.; Data curation, A.-Q.T. and S.-C.C.; Formal analysis, A.-Q.T., S.-C.C., J.-S.P., and Y.L.; Investigation, A.-Q.T.; Methodology, A.-Q.T., S.-C.C., J.-S.P., and Y.L.; Software, A.-Q.T.; Validation, J.-S.P.; Visualization, A.-Q.T. and S.-C.C.; Writing—original draft, A.-Q.T.; and Writing—review and editing, S.-C.C. and J.-S.P. All authors have read and agreed to the published version of the manuscript.

**Funding:** This paper was supported by National Natural Science Foundation of China with grant number NSF 61872085, Natural Science Foundation of Fujian Province with grant number 2018J01638, and project 2018Y3001 of Fujian Provincial Department of Science and Technology.

**Acknowledgments:** This paper was supported by National Natural Science Foundation of China with grant number NSF 61872085, Natural Science Foundation of Fujian Province with grant number 2018J01638, and project 2018Y3001 of Fujian Provincial Department of Science and Technology.

**Conflicts of Interest:** We wish to confirm that there are no known conflicts of interest and there has been no significant financial support for this work that could have influenced its outcome. We confirm that the manuscript has been read and approved by all named authors and that there are no other persons who satisfied the criteria for authorship but are not listed.

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
