# Peer review of "A Novel Pigeon-Inspired Optimization Based MPPT Technique for PV Systems"

_processes, doi:10.3390/pr8030356_

Round 1

Reviewer 1 Report

This interesting paper presents a novel Pigeon-Inspired Optimization technique that is used for MPPT in PV systems. PIO is a relatively new concept that has some advantages over traditional methods such as PSO, and as far as I am aware this is the first time it has been applied to MPPT in PV applications. Overall the paper is interesting and useful, but it is difficult to read and needs quite a lot of work to the structure. My main comments are:

  1. The introduction has large sections that talk about current optimisation techniques (lines 58 - 74) that I think could be removed as they are not directly relevant.
  2. The related work section contains circuit models of the PV cells, but these modes are not directly used in the new method (the method does not depend in some way on the specific form of the model). So this section could be vastly simplified or removed all-together.
  3. By "power tube" do the authors mean transistor or an actual thermionic emission valve? I think they mean transistor!
  4. Likewise, the description of the duty cycle equations for the boost converter are not really relevant to this paper and should be removed.
  5. The P&O section is important and should be expanded to include other approaches for MPPT in PV applications.
  6. A Dove is not the same thing as a Pidgeon. In scientific and ornithological practice, "dove" tends to be used for smaller species and "pigeon" for larger ones.
  7. It is never clearly explained what the map and compass operator are? The compass factor R, for example, is defined but never used.
  8. Equation 12 appears very similar to the conventional PSO algorithm, the key differences could be discussed.
  9. There are many undefined variables in the pseudo-code (such as m, and c), these need to be defined in the text.
  10. The important results use an existing test function, this function should be at least partially described using a paragraph, rather than asking the reading to read another paper.
  11. The important plots in Figure 6 are too small to see clearly when printed on A4 paper, I think you need to make these two figures at least so they are bigger.
  12. It is not clear which of the test functions is most like the MPPT problem?
  13. In section 6 it is not clear how the fitness was computed for each Pidgeon, as in real life this must be sequential? (i.e test one power point, then another, then another)
  14. The text in Figure 10 is also too small to read.
  15. The plots in Figure 12 have no axis labels
  16. The overall manuscript is readable but has some major grammatical mistakes and would benefit from a careful proof-read.

Author Response

Response to Reviewer Comments

Dear  Reviewer:

Thank you for your comments concerning our manuscript entitled“A Novel Pigeon-Inspired Optimization based MPPT Technique for PV Systems” (ID: processes-748050). Those comments are all valuable and very helpful for revising and improving our paper, as well as the important guiding significance to our research. We have studied comments carefully and have made correction which we hope meet with approval. Revised portions are marked in blue in the paper. The main corrections in the paper and the responds to the reviewer's comments are as follows:

Point 1: The introduction has large sections that talk about current optimization techniques (lines 58-74) that I think could be removed as they are not directly relevant.

Response 1: 

This article (lines58-74) has spent a lot of time introducing the development and introduction of intelligent algorithms. The purpose is to show the importance of intelligent algorithms in the development of computers. It also introduces several important algorithms for evolutionary computing and shows the importance of the new algorithm.

Point 2: The related work section contains circuit models of the PV cells, but these modes are not directly used in the new method (the method does not depend in some way on the specific form of the model). So this section could be vastly simplified or removed all-together.

Response 2: 

The PV cells model included in the relevant work section refers to a brief model of photovoltaic cells, which can be easily understood, and this model has been reflected in the photovoltaic circuit model in section 6 later. Some irrelevant parts have been deleted in related work (lines148- 161).

Point 3: By "power tube" do the authors mean transistor or an actual thermionic emission valve? I think they mean transistor!

Response 3: 

We are very sorry for our negligence. Thank you very much for review comments. The author did not notice the use of nouns in the article and has changed "power tube" to "transistor".

Point 4: Likewise, the description of the duty cycle equations for the boost converter are not really relevant to this paper and should be removed.

Response 4: 

The author has removed the relevant parts of the Boost circuit.

Point 5: The P&O section is important and should be expanded to include other approaches for MPPT in PV applications.

Response 5:

For the content of the P&O section, this article has not been introduced in more detail, so the content of the P&O is added in section 2.3.

Point 6: A Dove is not the same thing as a Pigeon. In scientific and ornithological practice, "dove" tends to be used for smaller species and "pigeon" for larger ones.

Response 6:

We are very sorry for our negligence.  The author has changed 'dove' to 'pigeon'.

Point 7: It is never clearly explained what the map and compass operator are? The compass factor R, for example, is defined but never used.

Response 7:

In the introduction in section 3, the tools leading the pigeon have been introduced. The map and compass operator are the magnetic field in the first stage and the sun in the pigeon's mind. The author added a sentence ’map and compass operator as abstract expression of magnetic field and sun angle in pigeon's mind’ . R is used in Equation 12.

Point 8: Equation 12 appears very similar to the conventional PSO algorithm, the key differences could be discussed.

Response 8:

The author adds a sentence near Equation 12. ‘The speed update formula of the PSO algorithm is: Vt+1 =w* Vt+c1*rand(0,1)(Pbest-X)+c2*rand(0,1)(Gbest-X), which is the migration direction of the simulated bird swarm. The inertial weight is introduced, and the inertial weight is adjusted linearly (or non-linearly) according to the process and the flight of the particles to balance the search. globality and speed of convergence. The speed update formula of PIO is:

V t+1= e-R*t*Vt+rand(0,1)(Gbest-X), which simulates the behavior of pigeons returning home. This formula is the speed update formula of the first stage. Unlike PSO, it has no individual optimal impact and has a strong global search ability to avoid the problem of getting into a local optimum.’

Point 9: There are many undefined variables in the pseudo-code (such as m, and c), these need to be defined in the text.

Response 9: 

In algorithm 1, m is the number of pigeons in each group, and c is 0.5. The explanation is given in the algorithm.

Point 10: The important results use an existing test function, this function should be at least partially described using a paragraph, rather than asking the reading to read another paper.

Response 10: 

Detailed information on each test function is given in section 5.

Point 11: The important plots in Figure 6 are too small to see clearly when printed on A4 paper, I think you need to make these two figures at least so they are bigger.

Response 11: 

For the problem that the picture in Figure 6 is too small, the author put 18 result pictures on two pages of A4 paper for readers to read.

Point 12: It is not clear which of the test functions is most like the MPPT problem?

Response 12: The function 21- 23 are most like the MPPT problem.

Point 13: In section 6 it is not clear how the fitness was computed for each Pigeon, as in real life this must be sequential? (i.e test one power point, then another, then another).

Response 13: We have added a formula for calculating fitness values for pigeons in section 6 to solve this problem.

Point 14: The text in Figure 10 is also too small to read.

Response 14: The author places Figure 10 on an A4 sheet for readers to read.

Point 15: The plots in Figure 12 have no axis labels

Response 15: The author has add axis labels to Figure 12.

Point 16: The overall manuscript is readable but has some major grammatical mistakes and would benefit from a careful proof-read.

Response 16:We are very sorry for our negligence. We have reviewed the paper and revised the grammar.

We appreciate for your warm work earnestly, and hope that the correction will meet with approval.​

Reviewer 2 Report

The article A Novel Pigeon-Inspired Optimization based MPPT
technique for PV systems is very well organized and written and aims to analysing the performance of two algorithms, namely Partical Swarm Algorithm (PSO) and improved pigeon algorithm. This objective is achived by first studying the mechanism of multi-peak output characteristics of photovoltaic arrays in complex environments, and then by proposing a multi-peak MPPT algorithm based on a combination of an improved pigeon population algorithm and an incremental conductivity method.

Though technical, I believe the research is very interesting to a wide range of readers and this is another strong point of the research besides the results that are being highlighted in the paper.

There are some minor typos and corrections that should be done, such as:

  • the Algorithm 1 from page 9 should be referred to from text;
  • eq 16 is not referred to from text, unless is a typo in line 211 where the author refer eq 14; please check and correct.

The authors should check the paper and avoid general affirmations such the one in line 319 ("In summary, PCPIO has greater competitiveness.") unless they have a strong justification for it.

I suggest authors to also point out the limitations they encountered during the research and to also give details about the software and architecture of computers used to run the tests.

Author Response

 Please add your reply to reviewers

Dear  Reviewer:

Thank you for your comments concerning our manuscript entitled“A Novel Pigeon-Inspired Optimization based MPPT Technique for PV Systems” (ID: processes-748050). Those comments are all valuable and very helpful for revising and improving our paper, as well as the important guiding significance to our research. We have studied comments carefully and have made correction which we hope meet with approval. Revised portions are marked in blue in the paper. The main corrections in the paper and the responds to the reviewer's comments are as follows:

Point 1: There are some minor typos and corrections that should be done, such as: the Algorithm 1 from page 9 should be referred to from text; eq 16 is not referred to from text, unless is a typo in line 211 where the author refer eq 14; please check and correct.

Response 1: 

We are very sorry for our negligence. Algorithm 1 on page 9 is referenced in Algorithm 6. The repetition of Eq 15 and Eq 16 has been corrected.

Point 2: The authors should check the paper and avoid general affirmations such the one in line 319 ("In summary, PCPIO has greater competitiveness.") unless they have a strong justification for it.

Response 2: 

The author added a sentence at the end of section5. ‘Comparing PCPIO with other algorithms in this paper, no matter in the convergence speed and the optimal value reached, more than half of the test functions analyzed from the experimental results show that PCPIO is quite competitive.’

Point 3: I suggest authors to also point out the limitations they encountered during the research and to also give details about the software and architecture of computers used to run the tests.

Response 3: 

Limitations in the process of the experiment Because the simulation method proves that the method works well, but the lack of real photovoltaic equipment to perform the experiment will have more parameters affecting the experimental results. The experiments performed in this article are performed on a system with Windows 10 and a CPU of i7-4710MQ 2.5GHZ. The software environment is MATLAB R2016b.

We appreciate for your warm work earnestly, and hope that the correction will meet with approval.​

Round 2

Reviewer 1 Report

Thank you to the authors who have addressed the majority of my comments and have significantly improved the manuscript. 

Reviewer 2 Report

The authors have addressed all my suggestions and thus I consider the paper can be published with Processes Journal.